# Response to Stimulations Inducing Circadian Rhythm in Human Induced Pluripotent Stem Cells

**DOI:** 10.3390/cells9030620

**Published:** 2020-03-04

**Authors:** Hitomi Kaneko, Taku Kaitsuka, Kazuhito Tomizawa

**Affiliations:** Department of Molecular Physiology, Faculty of Life Sciences, Kumamoto University, 1-1-1 Honjo, Chuo-ku, Kumamoto 860-8556, Japan; 188r5113@st.kumamoto-u.ac.jp

**Keywords:** induced pluripotent stem cells, circadian rhythm, clock genes, simulated body temperature, HIF-1α, hypoxic response

## Abstract

Regenerative medicine and disease modeling are expanding rapidly, through the development of human-induced pluripotent stem cells (hiPSCs). Many exogeneous supplements are often used for the directed differentiation of hiPSCs to specific lineages, such as chemicals and hormones. Some of these are known to synchronize the circadian clock, like forskolin (Frk) and dexamethasone (Dex); however, the response to these stimulations has not been fully elucidated for hiPSCs. In this study, we examined the response of clock genes to synchronizing stimulation, and compared it with fully differentiated cells, U2OS, and fibroblasts. The expression of clock genes did not show circadian rhythms in hiPSCs with Frk and Dex, which could be due to the significantly low levels of BMAL1. On the other hand, a circadian-like rhythm of D-box binding protein (*DBP*) expression was observed in hiPSCs by culturing them in an environment with a simulated body temperature. However, the inhibition of temperature-inducible factors, which are involved in temperature rhythm-induced synchronization, could not repress the expression of such rhythms, while the inhibition of HIF-1α significantly repressed them. In summary, we suggest that clock genes do not respond to the synchronizing agents in hiPSCs; instead, a unique circadian-like rhythm is induced by the temperature rhythm.

## 1. Introduction

Human-induced pluripotent stem cells (hiPSCs) are now an essential tool for drug development, disease modeling, and cell therapy [1,2]. However, the basic physiology of these cells has not been fully elucidated, especially the expression of circadian core clock machinery. It is important to clarify this machinery properly, when hiPSCs are used for differentiation for the following reasons. First, synchronizing agents are often used for specific differentiation. For instance, forskolin (Frk) is supplemented into a medium to induce naïve hiPSCs [3,4], while dexamethasone (Dex) is used for deriving mesenchymal stem cell-like cells and osteogenic, adipogenic, and chondrogenic differentiation [5,6]. Second, the circadian gene *clock* is known to regulate the cell cycle in mouse embryonic stem cells (mESCs) [7], and the state of the cell cycle affects the capacity for differentiation [8,9].

Circadian clocks are endogenous oscillators that exist at cellular levels and control 24-h physiological processes in organisms [10]. In each cell, the circadian clock is generated by a cell-autonomous transcriptional autoregulatory feedback loop by synchronizing sources of stimulation (light, hormone, temperature, etc.). The core clock genes include circadian locomotor output cycles kaput (CLOCK), and brain and muscle Arnt-like protein 1 (BMAL1), which encode activators, and period 1 (PER1), PER2, cryptochrome 1 (CRY1), and CRY2, which encode repressors [10]. The CLOCK-BMAL1 complex also activates the nuclear receptors REV-ERBα and REV-ERBβ (encoded by *NR1D1* and *NR1D2*, respectively), which compete at retinoic acid-related orphan receptor (ROR)-binding elements (ROREs) with RORα and RORβ (encoded by *RORA* and *RORB*, respectively) [11,12,13]. A third CLOCK-BMAL1-driven transcriptional loop involves the D-box binding protein (DBP), thyrotroph embryonic factor (TEF), and hepatic leukemia factor (HLF) [10]. In mammals, circadian clocks are found in all major organ systems of the body and various types of mammalian culture cells, including cancer cell lines and fibroblasts [10,14], and regulate many physiological processes, including body temperature, hormone secretion, glucose homeostasis, and cell-cycle progression [15].

While circadian clocks are generated in all major organs, previous studies have reported that the circadian rhythm of clock gene expression is not generated in mESCs [16,17,18]. These authors showed that circadian clock oscillation is induced upon differentiation. Furthermore, Lu et al. showed that knockout of the clock gene did not influence the maintenance of the pluripotent state in mESCs [7]. Based on these reports, it is theoretically believed that mouse pluripotent stem cells (PSCs) have no capacity to generate the circadian rhythm of clock gene expression and that this machinery is not essential for stem cell physiology. The mechanisms by which the generation of circadian clocks is repressed in mouse PSCs are relatively well understood [19]. In the case of human PSCs, Dierickx et al. showed that undifferentiated human ESCs do not possess circadian clock machinery [20], while Umemura et al. suggested that undifferentiated hiPSCs lack the circadian oscillator because of the posttranscriptional suppression of the CLOCK protein [21]. However, the critical mechanisms for the lack of a circadian oscillator in hiPSCs, and the response to other synchronizing stimulations have not been fully elucidated.

In this study, we investigated how hiPSCs respond to synchronizing stimulations in gene expression levels and why circadian clocks are not generated, even under such stimulations. As a circadian synchronizer, we used Frk, which activates cAMP signaling [22], and Dex, which binds to the glucocorticoid receptor and activates it. Furthermore, we tested gene expression under the environment of simulated body temperature, which elicits the circadian rhythm of clock gene expression in RAT1 fibroblasts [23]. Interestingly, this temperature rhythm in the culture elicited a circadian-like expression of the *DBP* gene, even in hiPSCs. Then, we investigated why such a circadian-like rhythm appeared in hiPSCs and found that another type of signaling was involved in that phenomenon, namely hypoxic signaling, which mediates this rhythm of gene expression in hiPSCs.

## 2. Materials and Methods

### 2.1. Cell Culture

U2OS, BJ cells, were obtained from ATCC (Manassas, VA, USA), and mouse embryonic fibroblasts (MEFs) prepared from the embryos of an ICR mouse. These cells were cultured in a high glucose Dulbecco’s modified Eagle medium (DMEM; Wako, Osaka, Japan) with 10% fetal bovine serum (FBS; Corning, Rochester, NY, USA) and 100 units/mL penicillin-0.1 mg/mL streptomycin (Thermo Fisher Scientific, Waltham, MA, USA) at 37 °C with 5% CO_2_. The hiPSC line 201B7 [24] was maintained under feeder-free conditions in a Cellartis^®^ DEF-CS™ 500 Culture System (Takara Bio, Shiga, Japan), according to the manufacturer’s instructions, at 37 °C with 5% CO_2_.

### 2.2. Stimulation to Synchronize Circadian Rhythm

The circadian rhythm was synchronized via stimulation of 0.1 µM Dex (Wako, Osaka, Japan) for 2 h or 10 µM Frk (Wako, Osaka, Japan) for 1 h. After each stimulation, the medium was replaced with a standard medium (referred as 0 h), and RNA extraction was performed every 4 h. The initial induction of PER1 was examined by adding 0.1 µM Dex or 10 µM Frk. RNA extraction was performed every 30 min, while the time of addition was 0 min. The simulated body temperature rhythm was achieved by using a 33 °C condition for 12 h and a 37 °C condition for 12 h, for a total of 5 days, and then the cells were cultured for another 24 h at 37 °C. RNA extraction was performed every 4 h from day 5 to day 6. Gene knockdown was performed using Dharmacon™ ON-TARGET***plus*** SMARTpool™ siRNA (Horizon Discovery, Lafayette, CO, USA), which consists of a mixture of four siRNAs for same target gene. Lipofectamine RNAiMAX (Thermo Fisher Scientific, Waltham, MA, USA) was used to add a final concentration of 20 nM siRNA. siRNA was added on day 4 of the temperature rhythm, and removed 24 h later. Hypoxic treatment was performed by culturing cells under a 1% O_2_ condition for the indicated periods in a multi-gas incubator (APM-30DR; ASTEC, Fukuoka, Japan).

### 2.3. Quantitative Real-Time PCR

Total RNA was extracted from cells using a TRIzol reagent (Thermo Fisher Scientific, Waltham, MA, USA) according to the manufacturer’s instructions. cDNA was synthesized using a SuperScript^®^ VILO™ cDNA Synthesis Kit (Thermo Fisher Scientific, Waltham, MA, USA) with random hexamers, according to the manufacturer’s instructions. Quantitative real-time PCR analysis was performed on a ViiA7 Real-Time PCR System (Thermo Fisher Scientific, Waltham, MA, USA) using the PowerUp SYBR Green Master Mix (Thermo Fisher Scientific, Waltham, MA, USA). mRNA expression was normalized by 18S or glyceraldehyde-3-phosphate dehydrogenase (GAPDH) expression. The primer sequences are shown in Table 1.

### 2.4. Western Blot Analysis

Cells were lysed using a RIPA buffer (50 mM Tris-HCl, pH 8.0, 150 mM NaCl, 1% NP-40, 0.5% sodium deoxy cholate, 0.5% SDS) with 1% protease inhibitor cocktail (Nacalai Tesque, Kyoto, Japan). When examining the phosphorylation level, a 1% phosphatase inhibitor cocktail (Nacalai Tesque, Kyoto, Japan) was added. After sonication, the samples were centrifuged at 20,000 g for 15 min. Laemmli’s sample buffer (0.38 M Tris-HCl, pH 6.8, 12% SDS, 30% β-mercaptoethanol, 10% glycerol, 0.05% bromophenol blue) was added to the supernatant and boiled at 95 °C for 5 min. When extracting the nuclear fraction and cytosolic fraction separately, NE-PER™ nuclear and cytoplasmic extraction reagents (Thermo Fisher Scientific, Waltham, MA, USA) were used according to the manufacturer’s instructions. Samples were separated by SDS-PAGE and transferred to a polyvinylidene difluoride (PVDF) membrane. Membranes were blocked with blocking one (Nacalai Tesque, Kyoto, Japan) and then incubated with an anti-glucocorticoid receptor (GR; Cell Signaling Technology, Danvers, MA, USA), anti-Phospho-CREB (Cell Signaling Technology, Danvers, MA, USA), anti-CREB (Cell Signaling Technology, Danvers, MA, USA), anti-BMAL1 (Cell Signaling Technology, Danvers, MA, USA), anti-CLOCK (Abcam, Cambridge, UK), anti-β-Actin (Medical and Biological Laboratories, Nagoya, Japan), or anti-histone H3 (Cell Signaling Technology, Danvers, MA, USA) overnight at 4 °C. Then, the membranes were washed and incubated for 1 h with a horseradish peroxidase-conjugated anti-rabbit IgG antibody (Thermo Fisher Scientific, Waltham, MA, USA) or an anti-mouse IgG antibody (Dako, Carpinteria, CA, USA). Then, the membranes were washed and incubated with an Amersham™ ECL™ prime western blotting detection reagent (GE Healthcare, Piscataway, NJ, USA). The proteins on the membrane were visualized with an ImageQuart400 (GE Healthcare, Piscataway, NJ, USA) device and quantified with the ImageJ software (National Institute of Health, Bethesda, MD, USA).

### 2.5. Flow Cytometry

For the cell cycle analysis, cells in the DNA synthesis phase were labeled with EdU using Click-iT™ Plus EdU Flow Cytometry assay kits (Thermo Fisher Scientific, Waltham, MA, USA), and the DNA was stained using a FxCycle™ PI/RNase staining solution (Thermo Fisher Scientific, Waltham, MA, USA). Then, 47 h after Frk or Dex stimulation, the EdU was added and labeled for 1 h. Cells were harvested using TrypLE Select (Thermo Fisher Scientific, Waltham, MA, USA), fixed with 4% paraformaldehyde, and conjugated with an Alexa Fluor ™ 488 dye to detect EdU. Then, PI staining was performed and analyzed on a FACSVerse flow cytometer (BD Biosciences, San Jose, CA, USA). For the evaluation of the pluripotency of hiPSCs, cells were harvested using TrypLE Select and resuspended in PBS containing 1% bovine serum albumin (BSA); then they were incubated with an anti-human SSEA-4 (Thermo Fisher Scientific, Waltham, MA, USA) or anti-mouse IgG3 isotype control (Thermo Fisher Scientific, Waltham, MA, USA) for 45 min on ice. Next, the cells were washed and resuspended in PBS containing 1% BSA. SSEA-4 positive cells were analyzed on a FACSVerse flow cytometer. Data were analyzed using the FACSuite software (BD, Biosciences, San Jose, CA, USA).

### 2.6. Microarray

The microarray analysis was performed on the total RNA samples using a TORAY 3D-gene oligo chip (TORAY, Tokyo, Japan) according to the manufacturer’s instructions. After culturing under the temperature rhythm for 6 days, the total RNA extracted at 8 h and 20 h was used. Genes that increased or decreased in their expression at 20 h were listed. A GO analysis using GeneCodis (http://genecodis.cnb.csic.es/analysis) was performed on significant genes.

### 2.7. Statistical Analyses

The data in all graphs are expressed as the mean ± standard error of the mean (SEM). A comparison between the two groups was analyzed by a two-tailed unpaired Student’s t-test. For multiple comparisons, a one-way analysis of variance (ANOVA) followed by a Dunnet post-hoc analysis was performed. A two-way repeated measures ANOVA was performed on the differences in the multiple time points of *DBP* expression between the siControl and target siRNA. The KaleidaGraph software (Version 4.1; Synergy Software, Reading, PA, USA) was used for all statistical analyses. A Cosinor analysis was performed for all the 24 or 48 h time series in order to determine the significant rhythmicity [25]. A *p* value of *p* < 0.05 was considered to be statistically significant.

## 3. Results

### 3.1. Dexamethasone (Dex) and Forskolin (Frk) Stimulation did not Elicit a Circadian Rhythm in hiPSCs

To investigate whether a circadian rhythm exists in undifferentiated hiPSCs, stimulation by Dex or Frk was performed. Dex stimulation synchronized the rhythm of the expression pattern of *PER2* and *BMAL1* in U2OS cells, a human bone cell line; the *PER2* rhythm was observed, but the *BMAL1* rhythm did not occur in hiPSCs (Figure 1A). Frk stimulation synchronized the circadian rhythm in the expression of *BMAL1* in BJ cells, a human fibroblast cell line, but significant rhythms were not observed in any genes of hiPSCs (Figure 1B). As previously reported, Dex and Frk stimulation induced a transient expression of *PER1* and then synchronized the circadian rhythm [26]. Therefore, we examined whether a transient *PER1* induction occurred in the hiPSCs. The expression of *PER1* increased 60, 90, and 120 min after Dex stimulation in the U2OS cells but was not induced in hiPSCs (Figure 1C). Similarly, the expression of *PER1* increased after the Frk stimulation in BJ and U2OS cells, but not in hiPSCs (Figure 1D). It has been reported that Clock is implicated in the cell cycle of mESCs [7]. Therefore, we examined whether Dex and Frk affect the cell cycle of hiPSCs. As a result, the percentage of the cells in each phase (G0/G1, S and G2/M) was not significantly changed by Dex and Frk stimulation (Figure 2).

To investigate why *PER1* could not be induced by Dex and Frk in hiPSCs, the mRNA and protein levels of downstream factors, and their responses to each stimulation, were examined (Figure 3 and Figure 4). The gene expression of *NR3C1*, which encodes the glucocorticoid receptor (GR), was lower in hiPSCs than in U2OS cells (Figure 3A). Then, responses to Dex were compared between hiPSC and U2OS cells using the nuclear translocation of GR as an index. The general GR levels in both the cytosol and nucleus were significantly lower in hiPSCs than in U2OS cells under a non-stimulated state. However, the ratio of nuclear/cytosolic GR levels was equivalent between the hiPSCs and the U2OS cells after Dex stimulation, showing that GR translocation occurred in hiPSCs, as well as in U2OS cells (Figure 3B,C). Next, the genes involved in cAMP–CREB signaling were compared between the hiPSCs and U2OS cells. The hiPSCs were expressed as usual in the major genes, such as *CREB1*, *CREB3*, *PRKACA*, *CREBBP*, *EP300,* and *CRTC2* (Figure 4A). Then, responses to Frk were assessed using CREB phosphorylation as an index. CREB phosphorylation normally occurs in hiPSCs, rather than BJ and U2OS cells (Figure 4B,C).

### 3.2. Characteristics of Clock Gene Expression in hiPSCs

As previously reported, CBP recruitment to the E-box in the promoter by BMAL1—but not by CREB—is crucial for the rapid induction of *PER1* [27]. Therefore, we measured and compared the steady state level of BMAL1 and other core clock genes. As a result, the levels of *BMAL1*, *PER1*, *PER2,* and *PER3* were significantly lower than the levels of U2OS cells (Figure 5A). Next, we examined the protein levels of BMAL1 and CLOCK and found that the BMAL1 protein level was extremely low in the hiPSCs (Figure 5B).

### 3.3. Circadian-Like Rhythms Occurred under a Circadian Temperature in hiPSCs

We attempted to clarify whether a circadian rhythm is elicited in hiPSCs when cultured under the simulated rhythms of body temperature, which is another synchronization factor for circadian rhythm. The human body temperature fluctuates between 36.5 °C and 37.5 °C, daily [28]. In the cultured cells, the circadian rhythm of clock gene expression was synchronized, even at a temperature fluctuation of 1 °C, while larger fluctuations like 4 °C synchronized it more efficiently [29]. We used the reliable protocol of 33 °C for 12 h and then 37 °C for 12 h to determine the presence or absence of the rhythms clearly. Consistent with the previous report, U2OS cells and mouse embryonic fibroblasts (MEFs) showed circadian rhythms in the genes expressed under this protocol (Figure 6A). Surprisingly, the circadian-like rhythm of *DBP* expression was observed in the hiPSCs. However, such rhythms were not observed in *PER2* and *BMAL1* expression (Figure 6A). To track this rhythm more clearly, *DBP* expression was measured at every 1 h interval in hiPSCs, and the data are shown in the Appendix A. Significant rhythmicity (*p* < 0.01) was observed in the *DBP* expression with an acrophase of 20.5 h. Next, we checked whether the circadian-like rhythm of *DBP* expression depends on the exit from pluripotency in hiPSCs under this protocol. As a result, the percentage of SSEA-4-positive cells was similar between the cells cultured at 37 °C constantly, or those at 33–37 °C after culturing for 6 days (Figure 6B), showing that hiPSCs did not exit from pluripotent state by culturing them under this temperature rhythm. 

### 3.4. The Circadian-Like Rhythm of DBP Expression was Not Abolished by the Inhibition of CIRBP, BMAL1, and HSF1

To investigate the mechanism through which the circadian-like rhythm of *DBP* expression was observed by the temperature rhythm, the genes *CIRBP*, *BMAL1,* and *HSF1* [29,30], which are known to be involved in temperature rhythm-induced circadian rhythm, were inhibited by siRNA in hiPSCs. Then, the dynamics of the target genes and *DBP* expression were analyzed. The expression of *CIRBP*, which codes the cold-inducible RNA-binding protein, showed the significant rhythmicity via the temperature rhythm. Its expression gradually increased when the cells were cultured at 33 °C; then, it began to decrease after switching the temperature from 33 to 37 °C. The overall expression of *CIRBP* was clearly inhibited by siCIRBP. *BMAL1* expression was unchanged by the temperature change, and *HSF1* slightly increased after switching the temperature from 33 to 37 °C. The expression of these two genes was inhibited by each siRNA. Altogether, the inhibition of these genes did not result in any significant changes in the circadian-like rhythm of *DBP* (Figure 7).

### 3.5. Hypoxia Signaling is Involved in the Circadian-Like Rhythm of Gene Expression by the Temperature Rhythm

A microarray analysis was performed on the hiPSCs at 8 h and 20 h after changing to 33 °C under the temperature rhythm, which was chosen because *DBP* expression was lowest at 8 h, and highest at 20 h. The gene expression levels were compared between 8 and 20 h. Then, upregulated (more than 1.8-fold) and downregulated genes (less than 0.56-fold) at 20 h, in comparison to 8 h, were compared (Figure 8A,B and Appendix A). As a result, 17 genes were upregulated (more than 1.8-fold), while 45 genes were downregulated (less than 0.56-fold). Next, a gene ontology (GO) analysis was performed on these genes, and found that annotation of the GO biological process GO:0001666 (response to hypoxia) was significantly enriched in both the up- and downregulated genes (Appendix A). Among these genes, we focused on *DDIT4* and *TFRC* because they have GO:0001666 (a response to hypoxia). Then, the dynamics of the levels of these genes were examined under the temperature rhythm. Similar to *DBP*, a circadian-like rhythm was observed in the expression of both *DDIT4* and *TFRC* (Figure 8C). Next, we checked the levels of the representative genes *VEGFA* and *EGLN1*, which are well known to be downstream genes of *HIF1A* and to show a response to hypoxia [31]. A circadian rhythm was also observed for *EGLN1* expression under the temperature rhythm (Figure 9).

### 3.6. The Circadian-Like Rhythm of DBP was Abolished by HIF-1α Inhibition

Next, we performed an inhibition of HIF-1α by *HIF1A* siRNA and measured the levels of *DBP* under the temperature rhythm. As a result, *HIF1A* mRNA levels were clearly repressed by siRNA, while the *DBP* rhythm was abolished in *HIF1A* siRNA-treated hiPSCs (*p* < 0.01 by two-way repeated measures ANOVA; Figure 10A). However, at a constant phase of 37 °C for 24 to 48 h, the repression of the *DBP* rhythm by HIF1A siRNA became weak but remained significant (*p* < 0.05 by two-way repeated measures ANOVA; Figure 10A). Then, we examined whether *DBP* expression is affected by hypoxia, and found that hypoxic treatment significantly increased its expression, similar to the known HIF-1α target gene, *VEGFA* (Figure 10B).

## 4. Discussion

In this study, we investigated whether the circadian clock is induced in hiPSCs by synchronizing stimulation. The results showed that the circadian rhythm was not induced by Dex and Frk in hiPSCs. Furthermore, we found that a circadian temperature in the culture elicited a circadian-like rhythm in the hiPSCs.

As shown in Figure 1, the circadian rhythm of *DBP*, *PER2,* and *BMAL1* expression was not observed after synchronizing stimulation (Dex and Frk) in the hiPSCs. Similarly, the circadian rhythm of *PER2* and *BMAL1* expression could not be identified in human embryonic stem cells (hESCs) after Frk stimulation [20]. Furthermore, we found that the early response of gene expression to Dex and Frk was repressed in hiPSCs compared to somatic cells (U2OS and BJ cells) (Figure 1C). The early induction of *PER1* and synchronization of the expression of clock genes by Dex was shown to be mediated by binding to the glucocorticoid receptor (GR) [32]. It is thought that activated GR translocates into the nucleus and then directly binds to the glucocorticoid response elements at the promoter of multiple clock genes, including *PER1* and *PER2* [33]. In this study, we showed that the levels of *NR3C1*, the mRNA of GR, were low in hiPSCs. Moreover, the accumulated GR protein levels in the nucleus were relatively low (even under Dex stimulation) compared to the U2OS cells (Figure 3), suggesting that *PER1* induction by Dex did not occur in hiPSCs due to low expression of GR. Frk is an activator of cAMP signaling and synchronizes circadian rhythms [22]. The primary action of Frk is the activation of protein kinase A (PKA); then, this kinase phosphorylates CREB [34,35]. The levels of *CREB1* and related genes were normal in hiPSCs, and the phosphorylation of CREB normally occurs in these cells (Figure 4). It has been reported that CBP recruitment to the E-box in the promoter by BMAL1, but not by CREB, is crucial for the rapid induction of *PER1* [27]. Accordingly, we tested the mRNA and protein levels of the main clock genes. As a result, the levels of the BMAL1 protein in hiPSCs were remarkably lower than U2OS cells (Figure 5B). Overall, it was suggested that the rapid induction of *PER1*, and the following circadian rhythm of clock gene expression, did not occur in hiPSCs, which could be due to the low expression of BMAL1. 

In hiPSCs, a circadian-like rhythm of *DBP* expression was observed under a circadian cycle of environmental temperature (33–37 °C) in the culture (Figure 6). The pluripotency of these cells was unaffected after culturing with this protocol, showing that this rhythm does not result from the differentiation. It has been revealed that the circadian rhythm of clock genes is induced by the temperature rhythm in the culture using RAT1 fibroblasts and that the clock genes regulate the induction of this rhythm, which HSF1 and CIRBP also participate in [23,29,30]. However, the knockdown of these proteins did not affect the circadian-like rhythm of *DBP* expression (Figure 7), suggesting that this rhythm is not a clock-regulated phenomenon. The microarray analysis preformed on cells between 33 and 37 °C captured a subset of up- and downregulated genes. Based on the GO analysis, the genes involved in the pathway for the response to hypoxia were assumed to be modulated by the temperature rhythm (Appendix A). When the other pathways were focused, in the upregulated genes, the following were observed: GO:0044419: interspecies interactions between organisms; GO:0016567: protein ubiquitination was significantly enriched; GO:0006412: translation; GO:0044267: cellular protein metabolic processes were significantly enriched in downregulated genes. Translational repression is a well-characterized mechanism underlying a variety of stresses, including heat stress, that save the cellular energy and prevent the synthesis of unwanted proteins [36]. This suggests that increasing the temperature from 33 (8 h) to 37 °C (20 h) led to the downregulation of genes that experience GO:0006412: translation. Alteration of the composition of the ubiquitin–proteasome pathway is also induced to avoid unfolded protein aggregation under environmental stresses like heat shock [37]. The genes annotated in GO:0016567 (protein ubiquitination) might be upregulated by this increase in temperature. Surprisingly, the *DBP* rhythm was interrupted when HIF-1α was inhibited by siRNA (Figure 10), suggesting that the changes in *DBP* expression under the temperature rhythm could be caused by HIF-1α. This made us think that the activity of HIF-1α is influenced by temperature changes. Supporting this idea, the levels of HIF-1α downstream genes were also affected by the circadian temperature (Figure 9). Hypoxia signaling features a feedback loop. In normoxic conditions, the HIF-1α protein is hydroxylated by the EGLN family from proline residues, and is then recognized by pVHL and degraded via the ubiquitin proteasome pathway [31]. When cells are exposed to hypoxic conditions, the oxygen-dependent EGLN prolyl hydroxylases become inactive, and then the HIF-1α protein is stabilized and stimulates transcription of the HIF-1α target genes [31,38,39]. Both *EGLN1* and *EGLN3* are also HIF-1α target genes, and their expression is accordingly induced by hypoxia, suggesting that the induction of these transcripts might serve as a form of negative feedback [31,40]. It is assumed that these feedback loops might cause a circadian-like rhythm of *DBP* expression in hiPSCs. In our study, it was found that the induction of the *DBP* gene was affected by hypoxic conditions (Figure 10B), suggesting that HIF-1α might be involved in *DBP* transcription. On the other hand, repression of the circadian-like rhythm of *DBP* by the *HIF1A* siRNA in a constant phase of 37 °C for 24 to 48 h became weak (Figure 10A). This result suggests that another molecular pathway could be implicated in this rhythm. Further studies are needed to confirm the molecular mechanisms under the circadian-like rhythms in hiPSCs.

## 5. Conclusions

We have shown that hiPSCs could not respond to synchronizing agents such as Dex and Frk. This might be due to insufficient levels of the specific receptor and core clock protein, BMAL1. However, further studies are needed for final conclusions to be drawn about the mechanism of the unresponsiveness in hiPSCs. The circadian-like rhythm of *DBP* expression in hiPSCs elicited by the temperature rhythm might be due to the feedback loop via HIF-1α, but not via heat or cold inducible proteins and clock genes. This study suggests the properties of the responses to synchronizing stimulations in hiPSCs, and their possible molecular mechanisms.

## Figures and Tables

**Figure 1 cells-09-00620-f001:**
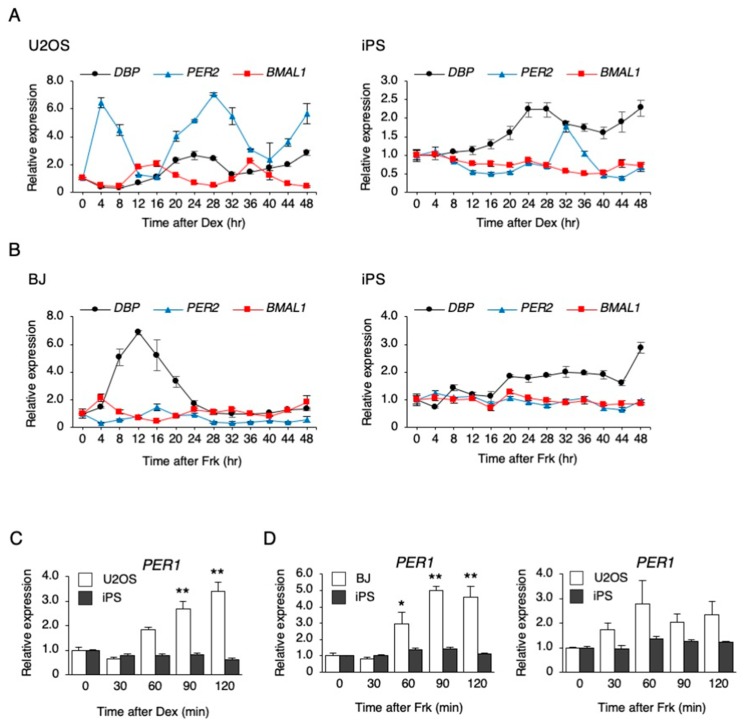
The response of clock genes to synchronizing stimulations in human-induced pluripotent stem cells (hiPSCs). (**A**) The expression levels of clock genes every 4 h after dexamethasone (Dex) stimulation in U2OS cells and hiPSCs. Data are presented as the means ± SEM relative to 0 h; n = 3 for each comparison. *P* values for the rhythmicity of *DBP*, *PER2*, and *BMAL1* were determined by a cosinor analysis as *p* = 0.08, *p* = 0.02, and *p* < 0.01 in U2OS cells, and *p* = 0.54, *p* = 0.03, and *p* = 0.11 in hiPSCs. (**B**) The expression levels of clock genes every 4 h after forskolin (Frk) stimulation in BJ cells and hiPSCs. Data are presented as the means ± SEM relative to 0 h; n = 3 for each comparison. *p* values for the rhythmicity of *DBP*, *PER2*, and *BMAL1* were determined by a cosinor analysis as *p* = 0.26, *p* = 0.29, and *p* = 0.02 in BJ cells, and *p* = 0.89, *p* = 0.21, and *p* = 0.59 in hiPSCs. (**C**) The expression level of *PER1* after treatment with 0.1 µM Dex in U2OS cells and hiPSCs. Data are presented as the means ± SEM relative to each 0 h. n = 3 for each comparison. ** *p* < 0.01 versus the corresponding control. (**D**) The expression level of *PER1* after treatment with 10 µM Frk in BJ, U2OS cells, and hiPSCs. Data are presented as the means ± SEM relative to each 0 h; n = 3 for each comparison. * *p* < 0.05, ** *p* < 0.01 versus the corresponding control.

**Figure 2 cells-09-00620-f002:**
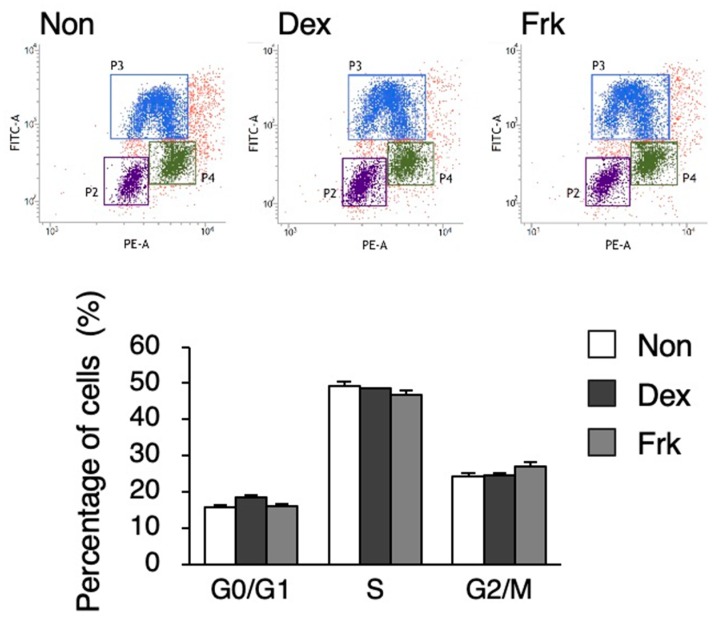
Cell cycle analysis after Dex and Frk stimulation in hiPSCs. Percentage of the cells in G0/G1; the S and G2/M phase in the hiPSCs after Dex or Frk stimulation is shown in the graph. Data are presented as the means ± SEM; n = 3 for each comparison.

**Figure 3 cells-09-00620-f003:**
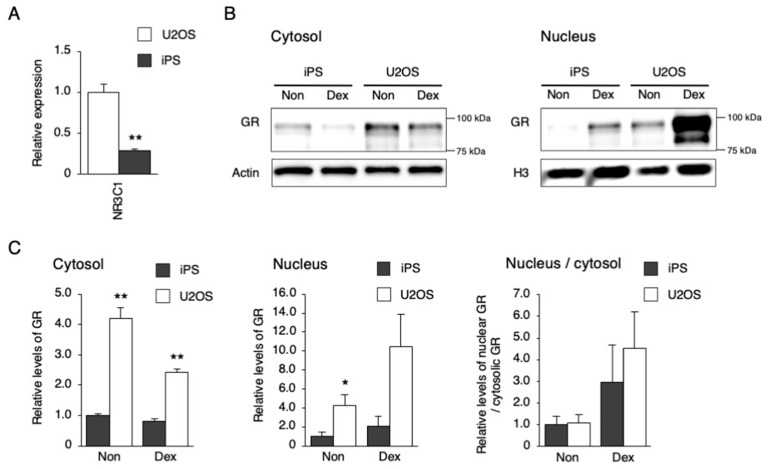
Response of downstream signaling to Dex stimulation in hiPSCs. (**A**) The expression level of *NR3C1* in U2OS cells and hiPSCs. Data are presented as the means ± SEM relative to U2OS; n = 4 for each comparison. ** *p* < 0.01 versus the corresponding control. (**B**) GR levels in the cytosol or nucleus after Dex stimulation. (**C**) Quantified GR levels in the cytosol or nucleus before and after Dex stimulation. Data are presented as the means ± SEM relative to hiPSCs; n = 3 for each comparison. * *p* < 0.05, ** *p* < 0.01 versus the corresponding control.

**Figure 4 cells-09-00620-f004:**
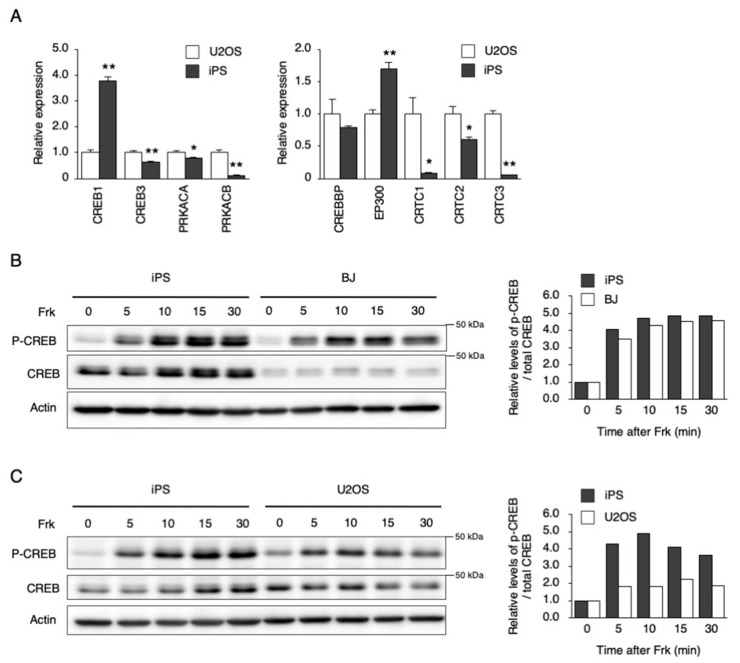
Response of downstream signaling to Frk stimulation in hiPSCs. (**A**) The expression levels of genes involved in cAMP signaling in U2OS cells and hiPSCs. Data are presented as the means ± SEM relative to U2OS; n = 4 for each comparison. * *p* < 0.05, ** *p* < 0.01 versus the corresponding control. (**B**) and (**C**) Phosphorylation of CREB by Frk stimulation in hiPSCs, BJ, and U2OS cells. Right graph shows quantified levels of p-CREB / total CREB from left blots, respectively. Data are presented as the means ± SEM relative to each 0 h.

**Figure 5 cells-09-00620-f005:**
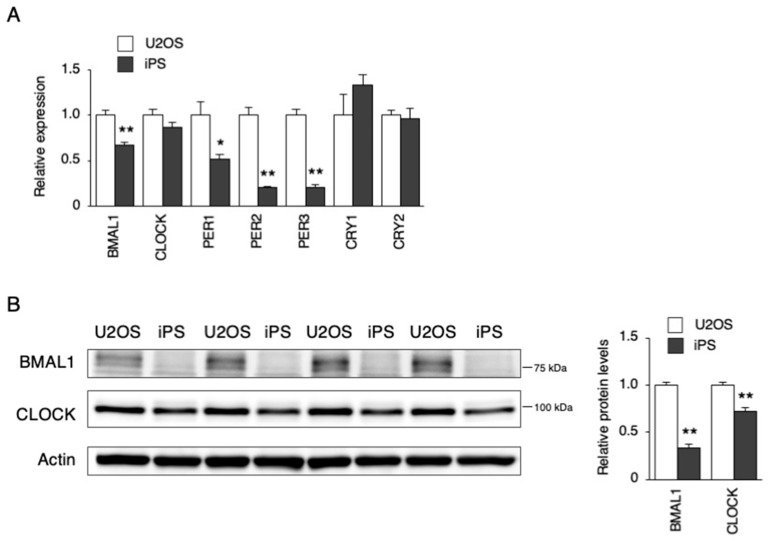
Expression levels of the clock genes in hiPSCs. (**A**) The expression levels of clock genes in U2OS cells and hiPSCs. Data are presented as the means ± SEM relative to U2OS; n = 4 for each comparison. * *p* < 0.05, ** *p* < 0.01 versus the corresponding control. (**B**) BMAL1 and CLOCK protein levels in U2OS cells and hiPSCs. Left, representative blot images. Right, the quantification of their blots. Data are presented as the means ± SEM relative to U2OS. n = 4 for each comparison. ** *p* < 0.01 versus the corresponding control.

**Figure 6 cells-09-00620-f006:**
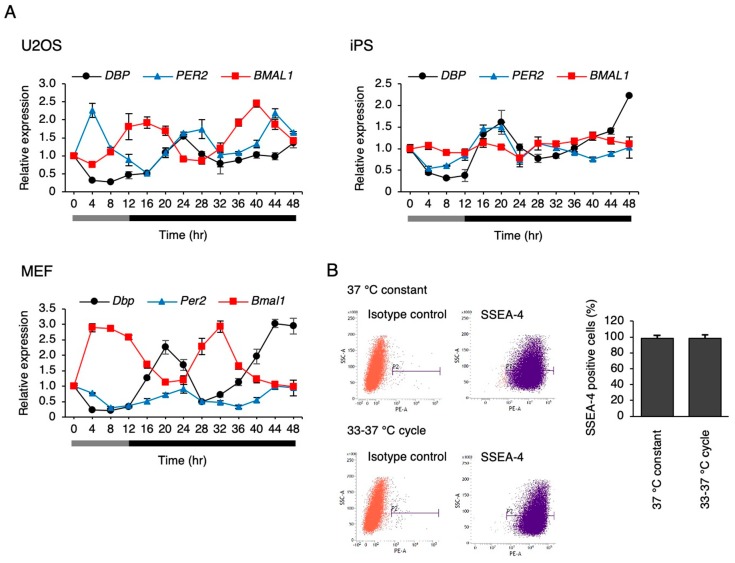
The response of clock genes to the simulated rhythm of body temperature. (**A**) The expression levels of clock genes under the temperature rhythm in U2OS cells, hiPSCs, and mouse embryonic fibroblasts (MEFs). 0 h refers to the start of day 5 of the temperature rhythm. The temperature was set at 33 °C from 0 to 12 h (gray bar) and then at 37 °C from 12 to 48 h (black bar). Data are presented as the means ± SEM relative to 0 h. n = 3 for each comparison. *P* values for the rhythmicity of *DBP*, *PER2*, and *BMAL1* were determined by a cosinor analysis with *p* = 0.04, *p* = 0.12, and *p* < 0.01 in U2OS cells; *p* = 0.02, *p* = 0.35, and *p* = 0.43 in hiPSCs; and *p* < 0.01, *p* < 0.01, and *p* < 0.01 in MEF. (**B**) The pluripotency marker SSEA-4-positive cells in hiPSCs after a 6 day culture according to temperature rhythm. P2 indicates the SSEA-4 positive region, and the bar graph shows the percentage of SSEA-4-positive cells. Data are presented as the means ± SEM; n = 4 for each comparison.

**Figure 7 cells-09-00620-f007:**
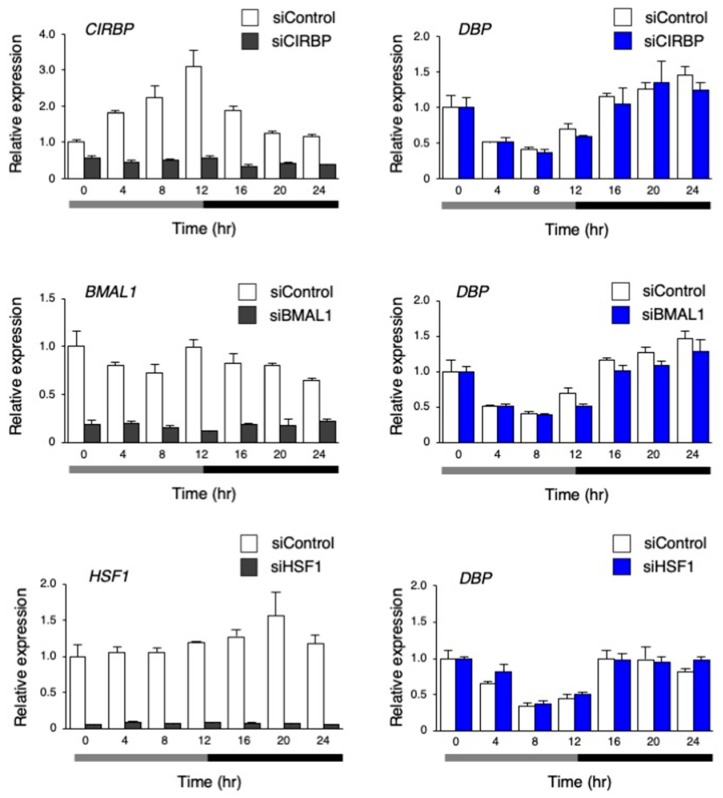
The effect of inhibiting the genes involved in the temperature rhythm-induced synchronization on the circadian-like rhythm of *DBP* expression. The expression levels of *CIRBP*, *BMAL1*, *HSF1,* and *DBP* under the temperature rhythm in hiPSCs treated with siControl, siCIRBP, siBMAL1, or siHSF1; 0 h refers the start of day 5 of the temperature rhythm. The temperature was set at 33 °C from 0 to 12 h (gray bar) and then at 37 °C from 12 to 24 h (black bar). Data are presented as the means ± SEM. The left graph shows the relative value to 0 h of siControl, and the right graph shows relative value with respect to 0 h for siControl and the target siRNA; n = 3 for each comparison. The *p* values for the rhythmicity of *CIRBP*, *BMAL1*, and *HSF1* in the siControl-treated hiPSCs were determined by a cosinor analysis with *p* < 0.01, *p* = 0.84, and *p* = 0.16, respectively.

**Figure 8 cells-09-00620-f008:**
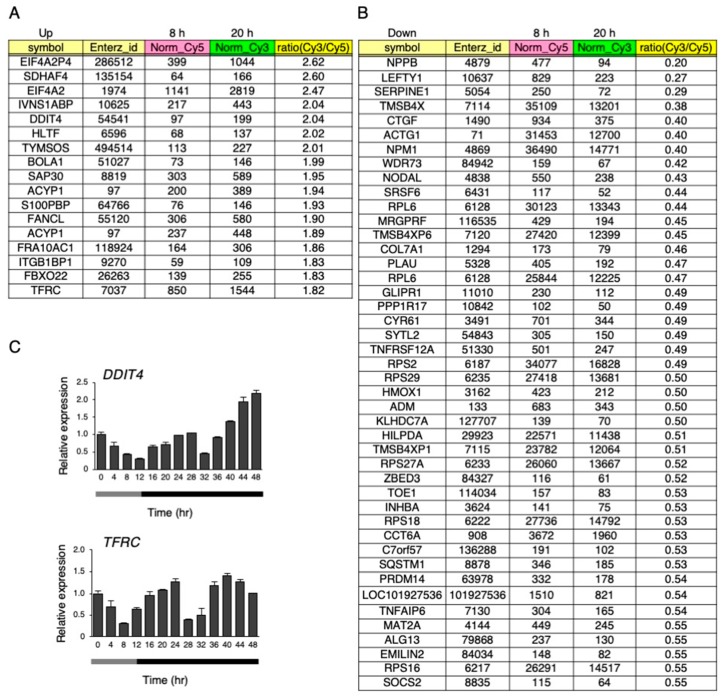
Up- or downregulated genes under the temperature rhythm. (**A**) and (**B**) Microarray analysis at 8 and 20 h on day 6 under the temperature rhythm. Total RNA mixed from three independent samples were used. Upregulated genes with a ratio (Cy3/Cy5) of more than 1.8-fold at 20 h are listed after the value of the normalized_Cy3 was cut off at more than 100 (**A**). Downregulated genes with a ratio (Cy3/Cy5) less than 0.56-fold at 20 h are listed after the value of the normalized_Cy5 was cut off at more than 100 (**B**). (**C**) The expression levels of *DDIT4* and *TFRC* under the temperature rhythm in hiPSCs; 0 h refers to the start of day 5 of the temperature rhythm. The temperature was set at 33 °C from 0 to 12 h (gray bar) and then at 37 °C from 12 to 48 h (black bar). Data are presented as the means ± SEM relative to 0 h; n = 3 for each comparison. The *p* values for the rhythmicity of *DDIT4* and *TFRC* were determined by a cosinor analysis with *p* = 0.07 and *p* < 0.01, respectively.

**Figure 9 cells-09-00620-f009:**
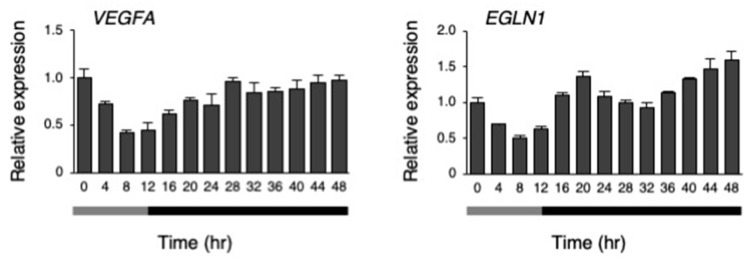
The response of hypoxia responsive genes to the temperature rhythm. The expression levels of *VEGFA* and *EGLN1* under their temperature rhythms in hiPSCs; 0 h refers to the start of day 5 of the temperature rhythm. The temperature was set at 33 °C from 0 to 12 h (gray bar) and then at 37 °C from 12 to 48 h (black bar). Data are presented as the means ± SEM relative to 0 h; n = 3 for each comparison. The *p* values for the rhythmicity of *VEGFA* and *EGLN1* were determined by a cosinor analysis with *p* = 0.18 and *p* < 0.01, respectively.

**Figure 10 cells-09-00620-f010:**
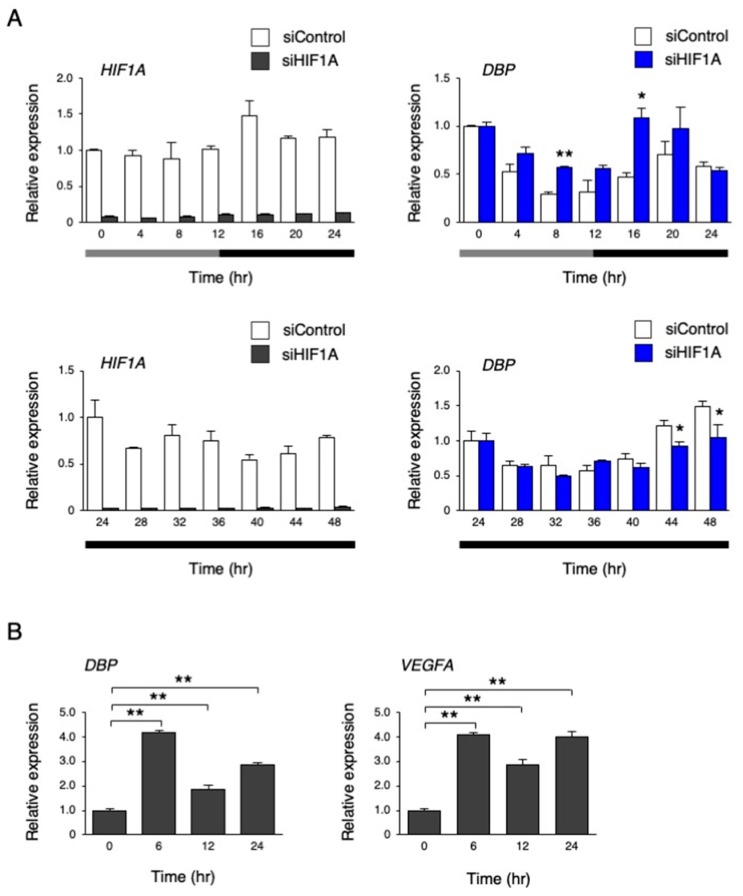
The effect of inhibiting HIF-1α on the circadian-like rhythm of *DBP* expression. (**A**) The expression levels of *HIF1A* and *DBP* under the temperature rhythm in hiPSCs treated with siControl or siHIF1A; 0 h refers to the start of day 5 of the temperature rhythm. The temperature was set at 33 °C from 0 to 12 h (gray bar) and then at 37 °C from 12 to 24 h and 24 to 48 h (black bar). Data are presented as the means ± SEM. The left graph shows the relative value to 0 h or 24 h of siControl, and the right graph shows the relative value for 0 h or 24 h, for siControl and the target siRNA; n = 3 for each comparison. A two-way repeated measure ANOVA revealed significant differences between siControl and siHIF1A (0 to 24 h, *p* < 0.01; 24 to 48 h, *p* = 0.02). Asterisks show the significant differences at each time point, with * *p* < 0.05, ** *p* < 0.01 (Student’s t-test). (**B**) The expression levels of *DBP* and *VEGFA* under a hypoxic condition (1% O_2_) in hiPSCs. Data are presented as the means ± SEM relative to 0 h; n = 3 for each comparison. ** *p* < 0.01 versus each value at 0 h.

**Table 1 cells-09-00620-t001:** Sequence of primers used for quantitative real-time PCR.

Species	Gene	Forward Primer	Reverse Primer
Human	18S	GAGGATGAGGTGGAACGTGT	GGACCTGGCTGTATTTTCCA
	GAPDH	TGTCAAGCTCATTTCCTGGTA	CACAGGGTACTTTATTGATGG
	DBP	CCAATCATGAAGAAGGCAAGAAA	GGCTGCCTCGTTGTTCTTGT
	PER2	GCTGGCCATCCACAAAAAGA	GCGAAACCGAATGGGAGAAT
	BMAL1	GAGAAGGTGGCCCAAAGAGG	GGAGGCGTACTCGTGATGTT
	NR3C1	AGCAGTGGAAGGTAGACAGC	CCTGTAGTGGCCTGCTGAAT
	CREB1	TTCTCCGGAACACAGATTTCA	AATCCTTGGCACTCCTGGTG
	CREB3	AGAGTGAGAGCTGTAGAAAAGAGG	AATCTTCCTCCGCACACGTT
	PRKACA	AAGAAGGGCAGCGAGCAG	CTGTGTTCTGAGCGGGACTT
	PRKACB	AGAGAACCACCTTGTAACCAGTA	TGGCTTTGGCTAGAAACTCTT
	CREBBP	TGGCTGAGAACTTGCTGGAC	TGGAAGCAGCATCTGGAACA
	EP300	ACCAGGAATGACTTCTAGTTTGA	TACGAGGCCCATAGCCCATA
	CRTC1	CAGCCGAGGCCAGTACTATG	AAGGGGGTCAGAGAGACAGG
	CRTC2	GGTGATGATGGACATCGGCT	CCGAGTGCTCCGAGATGAAT
	CRTC3	TGTGGGTTTTGACCAGCAGT	TCTTTGAACAGGCTGGTGCT
	CLOCK	ACGACGAGAACTTGGCATTG	TCCGAGAAGAGGCAGAAGG
	PER1	CCCAGCACCACTAAGCGTAAA	TGCTGACGGCGGATCTTT
	PER3	GCCTTACAAGCTGGTTTGCAA	CTGTGTCTATGGACCGTCCATTT
	CRY1	ACTCCCGTCTGTTTGTGATTCG	GCTGCGTCTCGTTCCTTTCC
	CRY2	TCTTCCAGCAGTTCTTCC	GTAGTCCACACCAATGATG
	CIRBP	CAGATCTCTGAAGTGGTGGT	CCTGCCTGGTCTACTCGGAT
	HSF1	TGCAGCTGATGAAGGGGAAG	AGGATCCGGTTTGACTGCAC
	DDIT4	GGTTCGCACACCCATTCAAG	CAGGGCGTTTGCTGATGAAC
	TFRC	GGACGCGCTAGTGTTCTTCT	CATCTACTTGCCGAGCCAGG
	VEGFA	CCCTGATGAGATCGAGTACAT	CGGCTTGTCACATCTGCAAGT
	EGLN1	AGCCCAGTTTGCTGACATTG	TCGTGCTCTCTCATCTGCATC
	HIF1A	CCTCTGGACTTGCCTTTCCT	TGGCTGCATCTCGAGACTTT
Mouse	18s	GAGGATGAGGTGGAGCGAGT	GAACCTGGCTGTACTTCCCA
	Dbp	CGTGGAGGTGCTTAATGACCTTT	CATGGCCTGGAATGCTTGA
	Per2	ATGCTCGCCATCCACAAGA	GCGGAATCGAATGGGAGAAT
	Bmal1	CCAAGAAAGTATGGACACAGACAAA	GCATTCTTGATCCTTCCTTGGT

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
