# Peer review of "Response to Stimulations Inducing Circadian Rhythm in Human Induced Pluripotent Stem Cells"

_cells, 2020, doi:10.3390/cells9030620_

Round 1
Reviewer 1 Report
Kaneko et al study the effect of known clock synchronizing agents in human induced pluripotent stem cells. They show that compared to differentiated cells, iPSCs do not manifest circadian oscillations in core clock gene expression upon forskolin or dexamethasone treatment, two commonly used drugs for manipulating pluripotency as well as differentiation. The authors found weak induced rhythmicity in DBP expression, when cells were synchronized in vitro with temperature cycles. The authors provide supporting data that this process depends on HIF1a signaling. Although the description of the absence of a core clock are highly in line with what has been published before in human embryonic stem cells (huESCs and hiPSCs), the authors do not describe any additional mechanistic insights on the absence of the circadian clock in pluripotent stem cells. Yet, the presence of HIF1a-based oscillations in iPSCs is highly interesting.
Major
-A huge flaw in this study is the complete lack of any statistical evidence that certain genes show circadian oscillations or not. Their cut-off for rhythmicity is objective and highly arbitrary. There are statistical tools out there to determine whether gene expression is circadian or not. The authors should at least report p-values for rhythmicity, based on RAIN, JTK cycle, or Cosinor analysis.
-The authors mention that the clock is connected to the cell cycle in mESCs. They should test whether forskolin and dexamethasone have an effect on the cell cycle. A basic EdU+PI FACs-based experiment must be done to rule out any cell cycle effects of these compounds.
-On line 180 the authors mention that nuclear translocation of GR upon dexamethasone is ‘much lower’ in hiPSCs compared to U2OS cells. There is no statistical quantification of the blots reported at all in the figure. In addition, the authors should calculate the ratio of cytosolic/nuclear GR abundance. If anything, it seems that in general the GR levels are lower in hiPSCs compared to U2OS cells, but that absolutely all of it translocates to the nucleus upon dexamethasone treatment.
-The authors should at least use a luciferase-based reporter system to track circadian rhythmicity of DBP over time.
-In figure 5, the authors depict clear circadian rhythmicity in CIRBP, which they completely omit to describe in the text. If CIRBP oscillates upon temperature cycles in hiPSCs it is a finding, and should be described properly in the text.
-In paragraph 3.5 the authors describe micro-array data of samples analyzed at 8h vs 20hs after changing temperature to 33degrees. The should include the whole micro-array results as a supplemental table. In the figure they only show 20 upregulated and 20 downregulated genes. What is the n? What is the statistical cut-off? The authors should perform gene ontology analysis on all the differentially expressed genes. In addition, the authors should at least describe why they zoom in on DDIT4 and TFRC.
Minor
-On line 36, the authors reference a paper that only looks at the clock gene ‘clock’, so therefore stating that clock genes (plural) are known to regulate the cell cycle in mESCs is overstated. If there are additional reports on different clock genes regulating mESC cell cycle, the authors should include these, otherwise they should only report about one clock gene (clock).
-Clock gene mRNA expression could be plotted as line graphs in the same graph. A hallmark of the canonical TTFL-based circadian clock is anti-phasic expression of BMAL1 and PER2. Plotting these genes as lines in the same graph would make the point stronger that there is a clear clock in differentiated cells, while iPSCs do not show this rhythmicity.
-In figure 1B, the authors should also plot the data for 48hrs (like they did) in figure 1A.
-In figure 3B, the authors should show replicate samples in the western blots instead of one vs one.
-In figure 5 and figure 8, the authors should depict data from at least 48hrs.
-On line 268 it should be EGLN1 instead of EGNL1.
Reviewer 2 Report
I enjoyed reading the manuscript of Kaneko et al and would like to congretulate them with their work. I think the article is interesting for the cells audience and recommend its publication
Some comments:
Previous studies showed that there is a large difference in the absolute expression of core clock components between differentiated and undifferentiated cells, including mRNA. Is this also true for the data in figure 1? Currently, only relative data is presented. In figure 1a, the authors observed rhythms in U2OS cells that are not present in iPS cells. How do the authors decide if data in a graph is rhythmic or not? (eg in the iPS DBP data, one could observe 2 peaks at 24 and 48 hours). Did they use statistics? If so, please provide the information in methods. If not, why did they not use statistics? Are the replicates in the figures biological or technical replicates? The sentence is line 179 'was occured' is not proper english, please correct.Author Response
Please see the attachment.

Reviewer 3 Report
This is an interesting paper based on literature findings that mESCs and hiSCs do not possess clock machinery, and that hiPSCs lacked the circadian oscillator because of the posttranscriptional suppression of CLOCK protein.
The authors claim that forskolin and dexamethasone, which activate canonical pathways to stimulate the clock machinery, do not elicit rhythmic expression of clock genes in hiSCs, but that temperature cycle in the culture elicited a circadian-like expression of the DBP gene, even in hiPSCs.
Major comments
1. 37 and 33oC oscillation does not simulate what occurs in humans…it is not clear what was the reasoning to choose these temperatures
2.
41 The core clock genes include CLOCK and BMAL1, which encode
42 activators, and PER1, PER2, CRY1, CRY2 and DBP, which encode repressors [10]
This leads the reader to a misunderstanding of the clock machinery: The core clock genes are CLOCK and BMAL1 and PER1, PER2, CRY1 and CRY2. A third loop involves DBP (D-box binding protein), TEF (thyrotroph embryonic factor) and HLF (hepatic leukemia factor)
73 and 1% penicillin–streptomycin
Please give concentrations of antibiotics
84 Gene
85 knockdown was performed using Dharmacon™ ON-TARGETplus siRNA (Horizon). Lipofectamine
86 RNAiMAX (Thermo Fisher Scientific) was used to add a final concentration of 20 nM siRNA
It is recommended 2 sequences of siRNA for same target
Which primers were used for cDNA synthesis? random hexamers or oligodTs? Please present primers’ sequences in a Table Please do centrifugation notation in g not rpm Tryple select is used to harvest adhered cells, not to dissociate in single cells…duplets and triplets are excluded during the flow cytometry procedure Circadian-like is not the appropriate qualification for the oscillation. Why not use a rhythm analysis such as Circawave or El Temps? Which statistical analysis was used in Figs 1A and B? All Western blots should be quantified (Fig 2) Protein ladder should appear in all Western images What is the protocol for differentiation?
As a result, the percentage of SSEA-4-positive cells was
similar between the cells cultured at 37 °C constantly, or 33–37 °C after culturing for 6 days (Figure235 4B), showing a pluripotent state of hiPSCs showed such circadian-like rhythms.
As far as I know one cannot obtain evidence of rhythm in the flow cytometer images
246 3.4. Circadian-like rhythm of DBP expression was not due to a known pathway
This is too strong an assertive. The authors did not test all known pathways
268….VEGFA and EGNL1, which are well
269 known to be downstream genes of HIF1A and show a response to hypoxia [26], and found that a 270 circadian-like rhythm was also observed in their expression under the temperature rhythm (Figure 7)
Needs statistical analysis to be able to claim that a rhythm exists
Results of hypoxia treatment appear in the manuscript without a prior description of protocolFig 8 A Asterisk is provided for all bars and in the legend it says that ** means different from the respective control. It does not seem to be true for all time points
301 (B) The expression levels of DBP and VEGFA under hypoxic condition in
302 hiPSCs. Data are presented as the means ± SEM relative to 0 h. n = 3 for each comparison. **p < 0.01
303 versus the corresponding control.
Which bars correspond to the controls?
The authors should discuss microarray data. The discussion is very poor in front of as many data as they have …needs improvement
because these cells did
354 not have enough levels of the specific receptor and core clock protein, BMAL1
The data are not sufficient for this conclusion
356.This study looks at the response of hiPSCs to synchronizing
357 stimulation, and their possible molecular mechanisms
This is not a conclusion; it should be the aim of the study
Please rewrite CONCLUSIONS
Round 2
Reviewer 1 Report
-Line 48: typo: should be BMAL1
-Figure 2. In the legend the authors describe p-values, while there are no asterisks in the figure? This is very confusing. It seems that there is a difference between treatments.
Reviewer 3 Report
The authors have made substantial changes following the referees` recommendations. The article is now suitable for publication in Cells.